# Targeting the Structural Integrity of Extracellular Vesicles via Nano Electrospray Gas-Phase Electrophoretic Mobility Molecular Analysis (nES GEMMA)

**DOI:** 10.3390/membranes12090872

**Published:** 2022-09-09

**Authors:** Stephanie Steinberger, Sobha Karuthedom George, Lucia Lauková, René Weiss, Carla Tripisciano, Martina Marchetti-Deschmann, Viktoria Weber, Günter Allmaier, Victor U. Weiss

**Affiliations:** 1Institute of Chemical Technologies and Analytics, TU Wien, 1060 Vienna, Austria; 2Center for Biomedical Technology, Department for Biomedical Research, University for Continuing Education Krems, 3500 Krems, Austria

**Keywords:** gas-phase electrophoresis, extracellular vesicles, exosomes, nES GEMMA, nES DMA, NTA

## Abstract

Extracellular vesicles (EVs) are in the scientific spotlight due to their potential application in the medical field, ranging from medical diagnosis to therapy. These applications rely on EV stability during isolation and purification—ideally, these steps should not impact vesicle integrity. In this context, we investigated EV stability and particle numbers via nano electrospray gas-phase electrophoretic mobility molecular analysis (nES GEMMA) and nanoparticle tracking analysis (NTA). In nES GEMMA, native, surface-dry analytes are separated in the gas-phase according to the particle size. Besides information on size and particle heterogeneity, particle number concentrations are obtained in accordance with recommendations of the European Commission for nanoparticle characterization (2011/696/EU, 18 October 2011). Likewise, and in contrast to NTA, nES GEMMA enables detection of co-purified proteins. On the other hand, NTA, yielding data on hydrodynamic size distributions, is able to relate particle concentrations, omitting electrolyte exchange (and resulting EV loss), which is prerequisite for nES GEMMA. Focusing on EVs of different origin, we compared vesicles concentrations and stability, especially after electrolyte exchange and size exclusion chromatography (SEC). Co-isolated proteins were detected in most samples, and the vesicle amount varied in dependence on the EV source. We found that depletion of co-purified proteins was achievable via SEC, but was associated with a loss of EVs and—most importantly—with decreased vesicle stability, as detected via a reduced nES GEMMA measurement repeatability. Ultimately, we propose the repeatability of nES GEMMA to yield information on EV stability, and, as a result, we propose that nES GEMMA can yield additional valuable information in EV research.

## 1. Introduction

Extracellular vesicles (EVs) are bio-nanoparticles consisting of a lipid bilayer encompassing an aqueous lumen, and are secreted by cells. Their manifold functions and contributions to physiological and pathological biological processes and their suggested roles in tissue regeneration offer a high potential for future application of these vesicles in the medical field. EVs have been proposed as biomarkers and therapeutic platforms [1,2,3,4,5,6]. Furthermore, in case of human origin, EVs are envisioned to enable more adapted vaccination strategies and the elucidation of virus infection processes in the human body [7,8,9,10,11].

Clinical and therapeutical applications require isolated vesicles of high purity. At the same time, the structural integrity of EVs should be preserved. High purity of EV preparations, however, is often achieved (I) at the cost of a significant loss of vesicles, which may be ascribed to EV adsorption to isolation/separation materials [12] and (II) a significant loss in the structural integrity of recovered EVs after excessive polishing.

The heterogeneity of EVs in function and size, roughly ranging from 30 to 1000 nm in diameter, [3,13] complicate their characterization, especially when co-purified material should be assessed at the same time. Current techniques for EV characterization include, but are not limited to flow cytometry, nanoparticle tracking analysis (NTA), and microscopic techniques [14,15,16,17]. Likewise, studies regarding the suitability of capillary electrophoresis for EV separation have been reported [18,19]. An increasing number of studies also focus on the development of new characterization and detection methods. In our research, we focused on gas-phase electrophoresis by means of a nano electrospray gas-phase electrophoretic mobility molecular analyzer (nES GEMMA). nES GEMMA enables particle number-based detection of size-separated analytes and thus offers the ability to detect small molecules next to larger ones in the size range between single digit nanometers up to around 200 nm surface-dry particle diameter (or even larger assemblies in dependence of the applied instrumentation). The technique is based on the separation of single-charged, surface-dry analytes in the gas phase according to the electrophoretic mobility (EM) diameter of particles, and was first described by Kaufman and colleagues [20]. In detail, following a nES process, analytes are dried and subjected to charge equilibration in a bipolar atmosphere induced e.g., by a ^210^Po α-particle source, soft X-ray radiation or an alternating bipolar corona-discharge [21,22,23]. The separation of polydisperse aerosolized particles subsequently occurs in a nano differential mobility analyzer (nDMA). There, particles are directed in a particle-free high laminar sheath flow of ambient air and an orthogonal tunable electric field. By variation of the electric field strength, monodisperse particle fractions are obtained at the exit slit of the nDMA. These particles then trigger nucleation in a supersaturated atmosphere of either n-butanol or water and are finally detected via laser-light scattering in an ultrafine condensation particle counter (CPC).

nES GEMMA is a number-based particle detection technique. Therefore, it is in agreement with the recommendation of the European Commission for characterization of material in the nanometer size range (2011/696/EU from 18 October 2011). Additionally, it is of note that nES GEMMA is also known under several other names—nES Differential Mobility Analyzer (nES DMA) [24,25], LiquiScan ES [26], Macro Ion Mobility Spectrometer (MacroIMS) [27], or Scanning Mobility Particle Sizer (SMPS) [28].

Previous studies have investigated the applicability of gas-phase electrophoresis for various bio-nanoparticles, including proteins [29], liposomes [30,31,32], viruses, and virus-like particles (VLPs) [33,34] as well as EVs [35]. However, for the latter analyte class either only nES GEMMA results for particles exceeding 20 nm were discussed in relation to orthogonal techniques in terms of vesicle detectability and size [35] or a special focus was put on proteinaceous sample components of EV containing samples, putatively co-purified from solution [36]. In our current study, we investigated the suitability and applicability of nES GEMMA for the characterization of EVs in an in-depth approach. Regarding potential applications of EVs, we focused our studies on several EV sources and their structural integrity after various sample preparation processes. We determined the repeatability of nES GEMMA measurements as an indicator for EV stability in solution and we assessed the influence of EV origin and purification steps on the determined particle numbers. The stability of EVs concerning storage in solution [37,38] and detergent effects [39] have already been addressed in other studies and thus were not part of our research.

## 2. Materials and Methods

### 2.1. Chemicals and Reagents

Phosphate-buffered saline (PBS, without Ca^2+^ and Mg^2+^) was obtained from Life Technologies (Paisley, UK) and ammonium acetate (≥99.99% trace metals basis) from Sigma-Aldrich (St. Louis, MO, USA). Ammonium hydroxide (28.2%) from Sigma-Aldrich was applied for pH adjustment. Tween 20 was obtained from Fluka (St. Louis, MO, USA). High-purity water was supplied by a MilliQ-System (18.2 MΩ cm resistivity at 25 °C; Millipore, Merck, Darmstadt, Germany).

### 2.2. EV Enrichment

EVs were obtained from various blood cell sources: monocytes, mesenchymal stem cells (MSCs), and red blood cells (RBCs). A schematic drawing of the respective EV isolation and measurement workflow is presented in Figure 1.

#### 2.2.1. Enrichment of Monocytic EVs (mEVs)

mEVs were isolated following the protocol of Tripisciano and colleagues [40]. Monocytic cells (THP-1; American Type Culture Collection), were grown in humidified atmosphere at 37 °C and 5% CO_2_ in supplemented RPMI medium containing 10% EV-depleted fetal bovine serum (FBS). When stable growth was reached, cells were harvested by centrifugation at 450× *g*. Subsequently, cells were washed, and 1 × 10^6^ cells/mL were inoculated into supplemented RPMI medium containing 10% human AB serum and grown for 4 h. Thereafter, the cell suspension was centrifuged at 450× *g* for 5 min to deplete cells, and the remaining supernatant was further centrifuged (1500× *g*, 15 min, 4 °C) to remove debris. The remaining supernatant was collected and immediately used for EV isolation.

#### 2.2.2. Enrichment of MSC-Derived EVs (mscEVs)

MSCs were cultured as described by Almeria and colleagues [41]. The use of human tissue was approved by the ethics committee of the Medical University Vienna, Austria (EK Nr. 957/2011, 30 January 2013), and all donors gave written consent. Human MSCs were isolated within 8 h after surgery as previously described. MSCs from three donors (aged 20–70) were cultivated in standard medium composed of minimum essential medium (MEM) alpha (Thermo Fisher Scientific, Waltham, MA, USA), 0.5% gentamycin (Lonza, Basel, Switzerland), 2.5% human platelet lysate (PL BioScience, Aachen, Germany; filtered through 0.2 μm filters), and 1 IU/mL heparin (Ratiopharm, Ulm, Germany) in humidified atmosphere at 37 °C, 5% CO_2_, and 21% or 5% O_2_. When stable growth was reached, 3 × 10^5^ cells/mL were inoculated into supplemented MEM alpha medium containing 10% AB serum and grown for 6 h. Thereafter, the cell suspension was centrifuged at 1500× *g*, 15 min, 4 °C to remove cells and debris. The supernatant was stored at −20 °C until EV isolation.

#### 2.2.3. Enrichment of EVs from Red Blood Cells (RBCs)

The protocol for the enrichment of RBC-derived EVs was modified from Kitka and colleagues [42]. EDTA anticoagulated blood was collected from healthy volunteers (3 × 6 mL) with informed consent by venipuncture without a tourniquet through a 21-gauge needle by use of a vacutainer system (Greiner Bio-One, Kremsmuenster, Austria). Cellular components were sedimented from whole blood by centrifugation at 2500× *g* for 10 min. Plasma and the white blood cell containing buffy coat were removed and RBCs were suspended in equal volume of saline solution and washed three times at 2500× *g* for 10 min each at 4 °C. After washing, RBCs were diluted with equal volume of PBS (pH 7.4) and were kept at 4 °C for 7 days. At the end of the incubation period, the RBCs were removed by centrifugation at 1500× *g* for 10 min followed by another centrifugation step at 2850× *g* for 30 min. The remaining supernatant was collected and immediately used for EV isolation.

### 2.3. Isolation of EVs

EVs from supernatants (mEVs, mscEV, rbcEVs) were collected by sequential centrifugation as previously described [40,43]: EVs were pelleted at 20,000× g (30 min, 4 °C) using a Sorvall Evolution RC ultracentrifuge, Rotor SS-34 (Termo Fisher Scientifc, Waltham, MA, USA). The resulting pellet was washed with sterile PBS, re-centrifuged at 20,000 × g (30 min, 4 °C), re-suspended in PBS to a final protein concentration of 4 mg/mL (DC protein assay, Bio-Rad, Hercules, CA, USA), aliquoted, and stored at −80 °C until further use (EV fraction I, containing mainly plasma-membrane derived microvesicles). The supernatant was centrifuged at 100,000× g (60 min, 4 °C), using an Optima MAX ultracentrifuge, MLA-80 Rotor (Beckman Coulter, Brea, CA, USA). The resulting pellet was washed in PBS and re-centrifuged, suspended at a protein concentration of 4 mg/mL, aliquoted, and stored at −80 °C until further use (EV fraction II, containing mainly exosomes). Only EV fraction I, enriched in phosphatidylserine-exposing vesicles derived from the plasma membrane, was used in this study. The detailed characterization of this fraction using nanoparticle tracking analysis, flow cytometry, imaging flow cytometry, as well as cryo-electron microscopy has been described in detail in a previous study [43].

### 2.4. Size Exclusion Chromatography (SEC)

The isolated samples were subjected to SEC purification after isolation. Aliquots of 500 µL of each sample used for purification with qEV columns (Izon Science, Burnside, Christchurch, New Zealand) and fractions, containing 500 µL each, were collected. Annexin5-positive (Anx5^+^) samples, determined with flow cytometry, as described below, were pooled and centrifuged at 20,000× *g* (30 min, 4 °C). The resulting pellets were resuspended in PBS yielding EV20k SEC fractions.

### 2.5. Flow Cytometric Characterization of Blood-Derived EVs

EVs were characterized by flow cytometry using a CytoFLEX LX device (Beckman Coulter, Brea, CA, USA) equipped with 405 nm, 488 nm, 561 nm, and 631 nm lasers. For staining, EV suspensions were diluted in sterile-filtered Annexin5 binding buffer (0.1 μm Minisart syringe filter, Sartorius Stedim Biotech, Goettingen, Germany) to a protein concentration of 1 µg/mL. Aliquots (100 μL each) of the diluted samples were incubated for 15 min at room temperature in the dark with APC or PE-conjugated Annexin5 (BD Biosciences, San Jose, CA, USA) as a marker for EVs exposing phosphatidylserine in combination with PB-conjugated anti-CD45 (Beckman Coulter) to detect monocytic EVs or with PE-conjugated anti-CD73 and with APC AF750-conjugated anti-CD90 (Beckman Coulter) to detect mesenchymal stem cell EVs or with APC-conjugated anti-CD235a to detect red blood cell EVs. To remove any precipitates, fluorochrome conjugates were centrifuged at 18,000× *g* for 10 min at 4 °C prior to use. Table 1 gives an overview on obtained data for EV characterization.

Calibration of the flow cytometer was performed with fluorescent silica beads (1 μm, 0.5 μm, 0.1 μm; excitation/emission 485/510 nm; Kisker Biotech, Steinfurt, Germany). The triggering signal for EVs was set to the violet side scatter (405 nm), and the EV gate was set below the 1 µm bead cloud as previously described [43,44,45]. Prior to analysis, stained samples were diluted 1:5 in sterile-filtered Annexin5 binding buffer (BD Biosciences, Franklin Lakes, NJ, USA). Acquisition was performed for 2 min at a flow rate of 10 µL/min. Data were analyzed using the Kaluza Software (version 2.1 S/N 834066318, Beckman Coulter). Appendix A demonstrate the calibration, isotype controls, and the gating strategy to characterize EVs in isolated EV fractions.

### 2.6. nES GEMMA Measurements 

For nES GEMMA measurements, a volatile electrolyte solution is required. Therefore, the original EV buffer, PBS, was exchanged to 40 mM ammonium acetate, pH 8.4 with 10 kDa molecular weight cut-off (MWCO) filters (Pall Laboratory, Port Washington, NY, USA) at 9300× *g* with an overall sample dilution of 1:10 (*v/v*). For nES, a pressure drop of 4.0 PSID (pound per square inch differential; ~0.28 bar) with an average of 2.0 kV spray voltage at the fused silica cone-tipped capillary (shaped in-house) with 25 µm inner diameter [46] in the nES aerosol generator (model 3080C, TSI Inc., Shoreview, MN, USA) was applied. The polydisperse aerosol was transported through a drying chamber with a gas flow of 1.0 L/min of dried, ambient, particle-free air, and 0.1 L/min CO_2_, where charge reduction with an alternating bipolar corona-discharge took place [23]. Subsequently, the single-charged, surface-dry particles were separated with a sheath flow of 8.0 L/min in a nano differential mobility analyzer (nDMA, model 3480C, TSI Inc.). After separation, the particles are subjected to nucleation in a n-butanol supersaturated atmosphere and detected by laser light scattering in an ultrafine condensation particle counter (CPC, model 3776C, TSI Inc.).

The measuring parameters and scanning time of 190 s with 20 s voltage re-setting allowed for a detection range of 3–91.4 nm electrophoretic mobility (EM) diameter. Five consecutive scans per sample were combined via their median to counteract possible unspecific detector events (electronic noise). Comparison of the individual consecutive scans allowed evaluation of nES GEMMA measurement repeatability. Subsequently, nES GEMMA spectra were plotted using the software Origin (version: 2020, OriginLab, Northhampton, MA, USA).

### 2.7. Nanoparticle Tracking Analysis (NTA)

The samples were diluted between 1:10,000 (*v/v*) and 1:1000 in high-purity water (MilliQ-System, 18.2 MΩ cm resistivity at 25 °C, Millipore, Merck, Darmstadt, Germany) or 40 mm ammonium acetate, pH 8.4 for measurement, depending on the origin of EV samples. All measurements were performed with a ZetaView PMX120 instrument (ParticleMetrix, Meerbusch, Germany) at 22 °C. The instrument determines the hydrodynamic particle diameter along 11 laser-light scattering measurement positions. The expected minimum brightness of the particles was set to auto, the particle size range to 5–200 nm, the shutter to 100, and the minimum tracelength and frame rate each to 15. The obtained data were analyzed with the corresponding instrument software (ZetaView 8.05.05 SP2, ParticleMetrix) to calculate median (X50), standard deviation, and size distribution, resulting in an average concentration of 10^11^–10^12^ particles/mL for EV samples and 10^8^–10^10^ particles/mL for SEC samples.

### 2.8. EV Disruption

rbcEV samples in ammonium acetate prior and after a SEC purification step were either treated with a Tween 20 solution (Fluka) to obtain a 0.05% Tween 20 concentration (Tween) in the sample or were subjected to ultrasonication (US) for 1 h in an ultrasonics bath (VWR, Radnor, PA, USA). Furthermore, the samples were also subjected to a combination of both treatments for a maximum impact on samples. Consecutively, all samples were measured via nES GEMMA and NTA.

## 3. Results and Discussion

In a previous study [47], we focused on one possible role of gas-phase electrophoresis on a nES GEMMA instrumentation for the characterization of EV containing samples. In doing so, we were able to demonstrate that gas-phase electrophoresis yields valuable information on sample quality regarding co-isolated proteinaceous components and hence, ultimately, information in terms of applicability of samples for their further in-depth characterization. With the current study, we focus on another possible role of gas-phase electrophoresis for EV characterization, namely, to gain information on vesicles themselves. In particular, EV stability and integrity were regarded, comparing isolation and purification steps as well as EV material of different cellular origin.

### 3.1. Sample Preparation of EV Material from Various Cellular Sources for Subsequent nES GEMMA Analysis

EVs from various blood cells and from a monocytic cell line were purified via ultracentrifugation applying standard protocols [42]. In order not to impair vesicle integrity due to high osmotic pressure gradients through membranes, preparation protocols often include final re-suspension of analytes in physiological electrolyte solutions, e.g., PBS. On the other hand, gas-phase electrophoresis relies on the transfer of bionanoparticle material from the liquid into the gas-phase in its native form via a nES process at atmospheric pressure. Therefore, all non-volatile sample components including solvent buffers are subsequently detected as analytes. Additionally, as particle number concentrations of non-volatile buffer components usually exceed those of actual analytes by far thus precluding sufficient sample dilution, non-specific particle aggregation of buffer components occurs during the electrospray process. Despite salt molecules being of significantly lower size than nanoparticle material in case of non-aggregated molecules, nES produced salt clusters impede the detection of actual bionanoparticle sample components due to higher EM diameter values of clusters, their heterogeneity, and high concentration [48]. Therefore, removal of original sample buffer components and concomitant exchange of the electrolyte to volatile ammonium acetate is a prerequisite of nES GEMMA. Concomitantly, we tried to keep the time of EVs in ammonium acetate prior analysis as short as possible to reduce the time of osmotic pressure exerted on EVs.

We opted for spin filtration on 10 kDA MWCO membranes in order to prepare samples for gas-phase electrophoresis. As analyte depletion on filter material due to analyte interactions with surfaces is a well-known phenomenon during nES GEMMA sample preparation [34], we analyzed our samples via NTA prior the electrolyte exchange step and at each intermediate step of preparation to evaluate respective EV loss. Figure 2 presents the resulting NTA data. Corresponding characteristics are summarized in Table 2. As shown, nES GEMMA sample preparation leads to a loss of vesicles, notably pronounced for mscEV, mEV, and rbcEV + SEC, while rbcEV seem to have a reduced interaction with the polyethersulfone spin filter material, resulting in a recovery of ≥50% vesicles. Possibly, high concentrations of co-purified proteins or vesicle stabilizing components (see below) might act as protective agents against EV surface interactions in this latter sample. All other EV sources showed an approximate reduction of vesicle numbers by 80%. At the same time, the hydrodynamic particle diameter of vesicles remained unaffected by nES GEMMA sample preparation as monitored by NTA.

### 3.2. Gas-Phase Electrophoresis of EV-Containing Samples

Following the NTA-based confirmation that samples still contained a sufficient amount of EV material after electrolyte exchange, we took our samples to nES GEMMA (Figure 3). Gas-phase electrophoresis at 8.0 L/min sheath flow inside the nDMA yielded signals up to 90 nm surface-dry particle diameter. In principle, a decrease of the nDMA sheath flow to 2.5 L/min allows detection of surface-dry particles even up to 190 nm EM diameter (Appendix A). However, no additional information on samples could be obtained for these latter nES GEMMA settings. Instead, the reduction of the sheath flow inside the nDMA from 8.0 to 2.5 L/min to scan a larger EM diameter range, impacted the peak width of homogeneous sample components (co-purified proteins). Hence, we opted for original instrumental settings (8.0 L/min sheath flow in the nDMA) as a compromise between the scanable EM diameter range (increasing with decreasing sheath flow) and broadening of peaks of homogeneous sample components of lower EM diameter (likewise increasing with decreasing sheath flow) for subsequent analyses.

For all EV containing samples, although of different origin, the overall protein pattern in the lower EM diameter range (<20 nm) did not show any significant difference in terms of particle EM diameter values. However, the number of detected particles was vastly different. Particularly for rbcEVs, counts reached 20,000 (Figure 3). It is of note that NTA failed to report these smaller-sized sample components [47]. Moreover, rbcEVs experiences least vesicle depletion during nES GEMMA sample preparation, possibly as a result of these co-purified small sample constituents (see below).

Furthermore, we detected bionanoparticle material exceeding 20 nm EM diameter—corresponding particle numbers, however, never exceeded 400 counts. We attributed these particles to a heterogeneous EV population. Fitting of Gaussian peaks to spectra (see Appendix A for an exemplary spectrum) resulted in peaks with characteristics as given in Table 1 (ST1). Several of these peaks exceeded 20 nm EM diameter and might thus correspond to EV subpopulations. Interestingly, in 2020, Brown and colleagues already reported various EV subpopulations in milk accessible via a home-built charge detection mass spectrometer (CDMS) [49].

### 3.3. Including Further EV Purification Steps in Sample Pretreatment

As demonstrated previously, application of an additional SEC step during EV purification significantly reduces the amount of co-purified material in the low EM diameter range as detected via gas-phase electrophoresis [47]. At the same time, NTA still reported the presence of vesicles. Focusing on analyte recovery and hydrodynamic particle diameter, we obtained results of our current study as reported in Figure 2 and Table 2, respectively. It is of note that for rbcEVs, particle recovery after SEC (rbcEVs + SEC) dropped significantly during nES GEMMA sample preparation in comparison to samples still containing high amounts of co-purified components (rbcEVs). In principle, the role of co-purified material might thus be two-fold: (I) inhibition of vesicle interactions to surfaces or (II) stabilization of EVs in solution. Next, in order to discriminate these two cases, we focused on vesicle stability.

For gas-phase electrophoresis, nES GEMMA results are usually obtained from several recorded spectra combined via their median to correct for few (if any) unspecific detector events recorded in individual measurements. However, especially for SEC purified EV containing samples we observed that the repeatability of consecutive measurements was considerably diminished in comparison to analytes being stable in solution (Figure 4, compare signals of rbcEVs + SEC and of rbcEVs). As a result, individual measurements appeared much noisier in case of highly purified vesicles than in samples still containing high amounts of co-purified material. From our experience, comparable effects can be recorded, when nES GEMMA capillaries are applied for liposome analysis and are dried, e.g., overnight, and re-flushed with ammonium acetate during a measurement series (unpublished data). In case of liposomes, we related the resulting noise to unspecific lipid fragments being released from the capillary surface after vesicle rupture upon drying and concomitant application of a shear force. In case of gas-phase electrophoresis of EV containing samples we also account resulting unspecific aggregates to the instability of lipid assemblies, i.e., EVs. However, in contrast to liposomes, we experienced signal instability for EVs already in solution. Morani et al. report similar effects for EV analyses via capillary electrophoresis [18]. As this effect was not observed for EVs without the additional SEC purification step, we finally reason that co-isolated proteins significantly increase the stability of EVs in solution. Regarding individual nES GEMMA spectra in addition to the resulting median of measurements therefore holds additional information on vesicle integrity and stability of these analytes.

To further investigate our finding, we subjected our samples to a treatment with detergents and ultrasonication. Similar to EV response to nES GEMMA sample preparation steps, NTA measurements related a decreased number in detected vesicles, particularly for a combinational treatment, although EVs were not completely removed from the samples (Figure 5). Especially, detergent treatment did not result in complete vesicle disruption. nES GEMMA at the same time, showed a reduced repeatability of consecutive spectra and an increased detection of unspecific aggregates as described already for SEC samples (Figure 6).

To conclude, our data provides evidence for the importance of investing additional efforts in the development of reproducible methods to enrich and purify native, intact EVs from complex matrices. Addition of well-defined smaller sample components such as proteins, mild detergents or similar might help to additionally stabilize vesicles in solution. Likewise, in-depth characterization of the obtained EV samples is highly advisable.

## 4. Conclusions

The stability of EVs is of importance for proper analysis of vesicle behavior and interactions with their surroundings. Various treatments, including pre-treatments for analysis techniques, influence the vesicle concentration, the hydrodynamic and surface-dry particle size, and the EV structural integrity. Especially for the latter, we found gas-phase electrophoresis on a nES GEMMA instrumentation to yield highly interesting data. Upon comparison of successive runs, we interpreted a reduced repeatability of spectra as a measure for reduced vesicle integrity. Co-purified proteins appeared to remedy this effect, most probably due to formation of a proteinaceous, non-covalently bound protein corona surrounding individual vesicles. Future approaches to purify and polish EV preparations might focus on the exchange of the ill-defined co-purified material by some more homogeneous compounds in order to increase EV stability in solution.

## Figures and Tables

**Figure 1 membranes-12-00872-f001:**
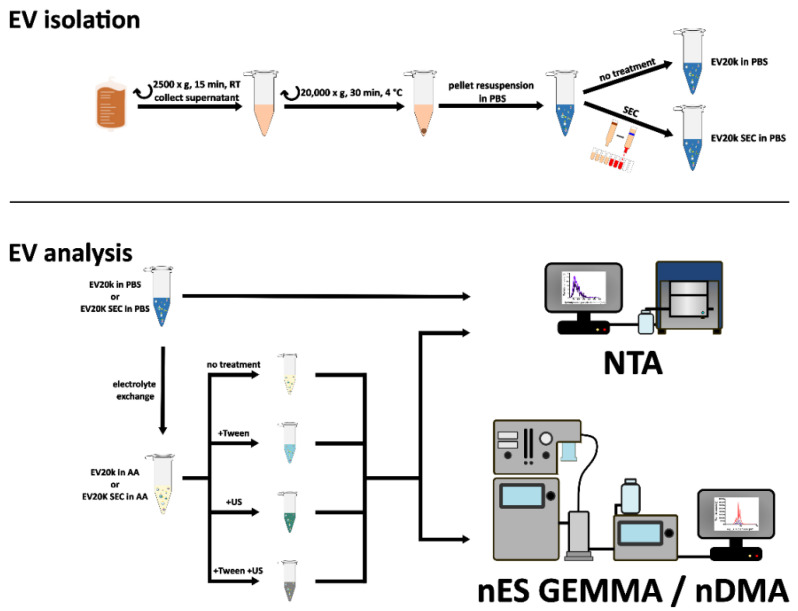
Schematic drawing of the workflow for extracellular vesicle (EV) isolation and EV analysis performed in this research work. The following abbreviations are used: AA—ammonium acetate, nES GEMMA—nano electrospray gas-phase electrophoretic mobility molecular analyzer, nES DMA—nano electrospray differential mobility analyzer, NTA—nanoparticle tracking analysis, PBS—phosphate-buffered saline, RT—room temperature, SEC—size exclusion chromatography, US—ultrasonication.

**Figure 2 membranes-12-00872-f002:**
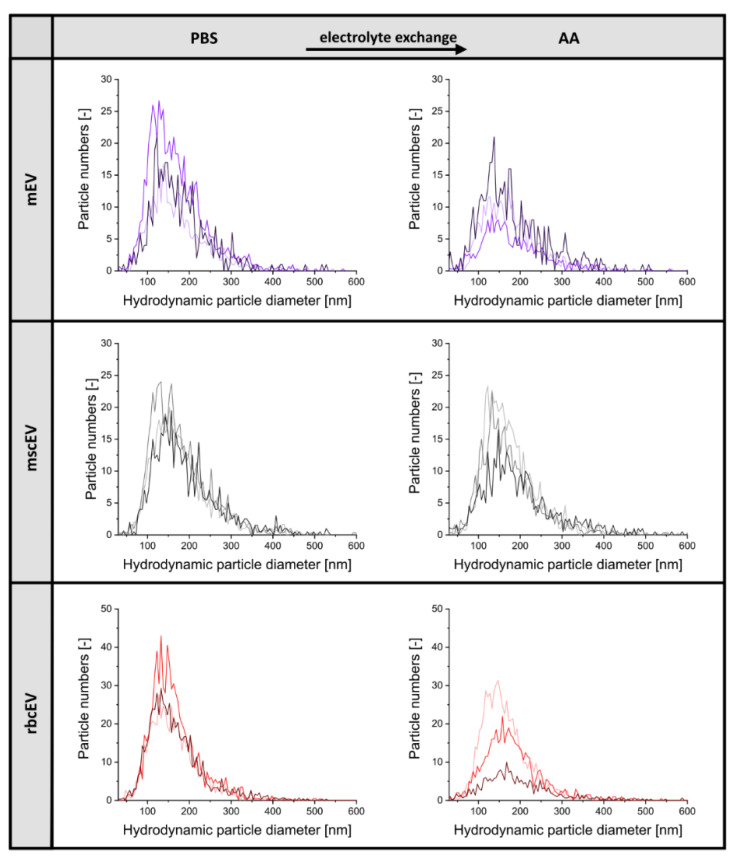
Comparison of EVs from different cell types (mEV, mscEV and rbcEV) and investigation of the impact of the electrolyte exchange from phosphate buffered saline (PBS) to ammonium acetate (AA) on the particle numbers and the hydrodynamic particle size distribution (NTA-derived).

**Figure 3 membranes-12-00872-f003:**
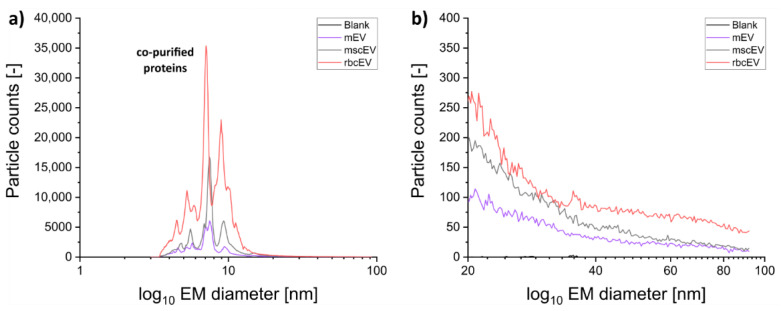
Characterization of EVs deriving from different cell types (mEV, mscEV, and rbcEV) and their surface-dry particle size distribution via nES GEMMA. The EV isolates reveal differences in the lower nm range in terms of detected particle numbers (co-purified proteins, (**a**)) in comparison to the upper nm range (**b**). There, changes in particle numbers are significantly less pronounced.

**Figure 4 membranes-12-00872-f004:**
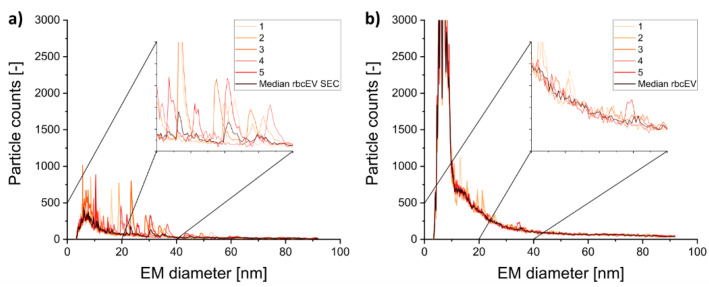
Comparison of nES GEMMA spectra of five single measurements and their median of rbcEV ± SEC purification, respectively. Measurements of rbcEV + SEC (**a**) reveal a significantly increased signal noise in contrast to rbcEV (**b**).

**Figure 5 membranes-12-00872-f005:**
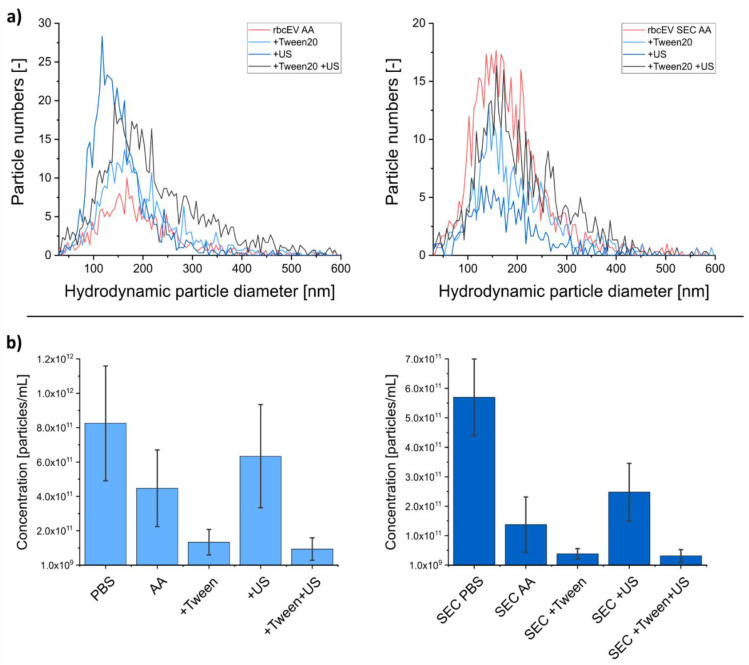
Influence of different treatments (Tween 20, ultrasonication (US)) on rbcEV isolates with/without size exclusion chromatography (SEC). Particle numbers and hydrodynamic particle diameters are regarded. The hydrodynamic particle diameter (**a**) of samples remains stable upon sample treatment. The impact on the particle concentration (**b**) shows loss of particle numbers after Tween 20 and ultrasonic treatments.

**Figure 6 membranes-12-00872-f006:**
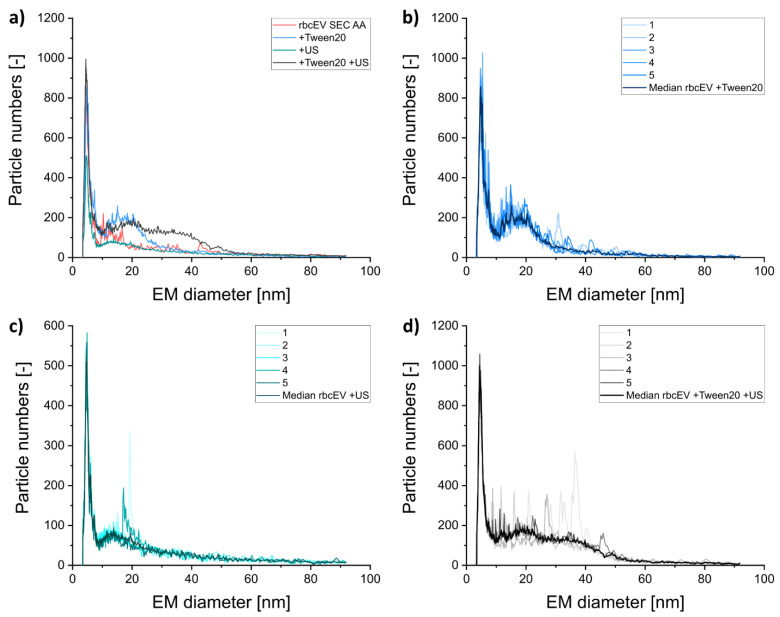
Relation of nES GEMMA spectra of rbcEV + SEC and the impact of different treatments (**a**) on the EV constitution and dry particle size diameter. The repeatability of the measurements for the treatments with Tween 20 (**b**), ultrasonication (**c**), and the combination (**d**) display the vesicle integrity.

**Table 1 membranes-12-00872-t001:** Characterization of applied EV preparations.

Sample	Protein Concentration (µg/mL)	EV Concentration (EVs/µL)	CD90^+^ EVs (% of all EVs)	CD73^+^ EVs (% of all EVs)	CD90^+^ CD73^+^ EVs (% of all EVs)
**mscEVs**	846 ± 117	1.64 ± 0.61 × 10^6^	46 ± 3	31 ± 9	27 ± 7
**Sample**	Protein concentration (µg/mL)	EV concentration (EVs/µL)	CD45^+^ EVs (% of all EVs)		
**mEVs**	864 ± 176	2.15 ± 2.19 × 10^6^	41 ± 6		
**Sample**	Protein concentration (µg/mL)	EV concentration (EVs/µL)	CD235a^+^ EVs (% of all EVs)		
**rbcEVs**	2486 ± 1129	4.80 ± 0.50 × 10^6^	83 ± 6		

**Table 2 membranes-12-00872-t002:** Comparison of EVs derived from different cell types (mEV, mscEV, and rbcEV) and investigation of the impact of solvent exchange from PBS to ammonium acetate (AA), as well as Tween20 and ultrasonic (US) treatment on the particle concentration and the hydrodynamic particle diameter (NTA-derived; *n* ≥ 7). The recovery is based on the particle concentration of the original sample in PBS compared to the particle concentration of the treated sample.

Sample	Concentration (Particles/mL)	Hydrodynamic Particle Diameter (nm) X50	Recovery (%)
**mscEV PBS**	4.35 ± 2.25 × 10^11^	163.7 ± 13.6	-
**mscEV AA**	7.63 ± 6.16 × 10^10^	162.5 ± 16.1	17.5
**mEV PBS**	4.88 ± 1.77 × 10^11^	156.1 ± 14.8	-
**mEV AA**	1.03 ± 0.61 × 10^11^	162.3 ± 19.5	21.1
**rbcEV PBS**	8.26 ± 3.34 × 10^11^	152.8 ± 12.1	-
**rbcEV AA**	4.48 ± 2.24 × 10^11^	163.1 ± 22.1	54.3
**rbcEV + Tween**	1.34 ± 0.74 × 10^11^	175.7 ± 28.8	16.3
**rbcEV + US**	6.35 ± 3.00 × 10^11^	142.1 ± 14.5	76.9 *
**rbcEV + Tween +US**	9.38 ± 6.58 × 10^10^	205.4 ± 37.1	11.4
**rbcEV SEC PBS**	5.70 ± 1.30 × 10^11^	155.7 ± 9.3	-
**rbcEV SEC AA**	1.38 ± 0.94 × 10^11^	166.3 ± 12.8	24.3
**rbcEV SEC + Tween**	3.95 ± 1.75 × 10^10^	177.9 ± 23.4	6.9
**rbcEV SEC + US**	2.49 ± 0.98 × 10^11^	158.4 ± 24.2	43.7 *
**rbcEV SEC + Tween + US**	3.22 ± 2.11 × 10^10^	199.1 ± 37.6	5.6

* Samples undergoing ultrasonic treatment display higher recovery rates, likely resulting from the creation of microbubbles or decomposition of unspecific, non-covalent aggregates > 600 nm.

## Data Availability

The data presented in this study are available on request from the corresponding author.

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
