# Peer review of "Targeting the Structural Integrity of Extracellular Vesicles via Nano Electrospray Gas-Phase Electrophoretic Mobility Molecular Analysis (nES GEMMA)"

_membranes, 2022, doi:10.3390/membranes12090872_

Round 1

Reviewer 1 Report

The authors describe the use of a novel method for the analysis of extracellular vesicles. The results may be interesting for the other researchers. I suggest the authors make some changes in the manuscript to improve the contents

1) Move the Fig S1 to the text of the manuscript

2) Add any information related to the nature of the EV, isolated as in Method 2.3 according to the MISEV 2018 - Transmission EM, cryo-EM. Describe the EV markers. Since the EV precipitation was made at 20000g, that is significantly lower than required for the exosome isolation. 

The authors analyzed several markers: Annexin5, CD45, CD73, CD90, and CD235a by flow cytometry. But the results are not presented in the paper. Also, the question is whether the sizes of EVs are enough to be visualized by flow cytometry on CytoFLEX LX?

3) The authors describe the detection of co-purifying proteins, but it is not obvious which fragment of which Figure shows these contaminants.

Since nothing is known about the size and structure of EVs, the sentence "we propose that nES GEMMA is a valuable tool in EV research" seems insufficiently substantiated.  

Reviewer 2 Report

It is an interesting and useful work presented in the manuscript and will be impactful in the field of EV research. 

1) The first concern is related to isolation EVs. Trying to understand how you were able to pellet EVs with only 20,000xg. To pellet EVs usually requires forces up to 100,000xg. 20,000xg usually allows to only obtain microvesicles. However, in the report you show that according to NTA and nES GEMMA you obtained EVs with hydrodynamic sizes of 100-300 nm which would require significantly higher forces to pellet them. Please explain this part. 

2) Please provide flow cytometry results. I only found information in the methods section. 

3) To confirm presence of EVs it is of common practice to image them after isolation to confirm successful isolation. Is it also possible to deposit them for SEM imaging by using nES GEMMA? Overall it would be good to have some EM images in the paper/SI.

Minor spell checks are also needed (e.g. Cytrometric instead  of Cytometric on page 4).

Round 2

Reviewer 1 Report

The manuscript in the current state doesnt contain any visualization of the EV's authors are analyzing. To my opinion, it is a critical issue. According to the MISEV2018, data obtained with TEM or other similar methods should be provided for every experiment, or fraction isolated. 

The authors cannot say that the "method demonstrated the integrity of EVs contained in the 20,000 g fraction was used in this study" - this is not evidence and this is not enough. 
